# A Framework for Implementing and Tracking Circular Economy in Cities: The Case of Porto

**António Cavaleiro de Ferreira** [1,*] and **Francesco Fuso-Nerini** [1,2]

1  Unit of Energy Systems Analysis (dESA), KTH Royal Institute of Technology, Brinellvägen 68, SE-100 44 Stockholm, Sweden; francesco.fusonerini@energy.kth.se
2  Payne Institute, Colorado School of Mines, Golden, CO 80401, USA
*  Correspondence: antonio57_cf@hotmail.com

**Abstract:** Circular economy (CE) is an emerging concept that contrasts the linear economic system. This concept is particularly relevant for cities, currently hosting approximately 50% of the world's population. Research gaps in the analysis and implementation of circular economy in cities are a significant barrier to its implementation. This paper presents a multi-sectorial and macro-meso level framework to monitor (and set goals for) circular economy implementation in cities. Based on literature and case studies, it encompasses CE key concepts, such as flexibility, modularity, and transparency. It is structured to include all sectors in which circular economy could be adopted in a city. The framework is then tested in Porto, Portugal, monitoring the circularity of the city and considering its different sectors.

**Keywords:** circular economy; urbanization; framework; indicators; circular city

## 1. Introduction

Circular economy (CE) is an emerging concept which is seen as the alternative to the current linear economy [1]. Its impacts are seen as relevant—now more than ever. Given the present environmental crisis, alongside economic uncertainty, governments have started seeking alternatives. China has made successful use of CE to tackle urban pollution and a wasteful system [2]. Its results have also motivated the European Union (EU) to promote it.

The EU has a variety of international environmental, technological, economical, and social goals. Particularly, it looks to follow the United Nations (UN) 17 Sustainable Development Goals (SDGs). A CE can be the answer to a systemic approach that eases the completion of several goals, specifically Goal 12 [3].

Alongside a CE comes urbanization, so the need to define and understand circular cities. This gives rise to a new set of research gaps, as the definition of a circular city, how to monitor the city's circularity, how to implement this circularity, which indicators to use, and what data to collect [4]. This paper explores these different research gaps, with a focus on developing a tool to measure and set goals of circularity in a city.

With this tool, the Circular City Analysis Framework (CCAF), the main goal is to help municipalities and cooperating actors understand a city's circularity. It is based on the city's intrinsic properties and sectors, as well as the circular economy and the circular city's characteristics. Moreover, it is structured to adapt to different cities, to other tools and future indicators. It was tested in the Portuguese city of Porto, as a case study.

The introduction of this paper is subdivided into the present global context, followed by CE key concepts. This is then bridged to the circular city definition, which is essential to the framework development. Finally, the Portuguese context is introduced, together with the northern city of Porto.

Second, it explores the methodology used to develop the framework. This section is subdivided into a literature review timeline, followed by a description of the framework's general characteristics. Relevance of the indicators of the methodology is explained, and a field-by-field description follows.

The case study of Porto is the Section 3, where the framework is implemented to monitor and reflect Porto's circularity. This is followed by the Section 4 where the case study results and the framework functionality are further discussed.

The paper ends with a conclusion. Extra information can be found in the appendixes, such as the complementary tables of the framework and the table of interviewed experts.

## 1.1. International Contextualization for Cicular Economy

Internationally, there is a notable effort to tackle environmental challenges such as climate change and air pollution. Technological disruptions and shifts are also emerging, mainly in the form of digitalization and new power and storage sources. These new technologies can be used to tackle the issues in innovative ways, allowing for ambitious goals.

Many countries embrace these goals, believing that the shift towards a more sustainable system can empower a country, while increasing the lifestyle of their areas and reducing international environmental stresses.

One of the most relevant sets of goals is that of the SDGs. They encompass different areas, acknowledging the different possible technologies and trends of the 21st century. These goals have holistic influence, focusing on social, environmental and economical progression, and are highly interconnected [5].

The EU also embraced a more sustainable path, reflected in several goals to reduce materials and energy use. Its goals include, by 2030, increasing municipal waste prepared to reused and recycling to 65%, packaging waste to reuse and recycling to 75%, limit municipal landfill waste to 10% and reduce marine litter by 30% [6]. Furthermore, the EU strives to meet global environmental goals, such as the below 2 °C target embedded in the Paris agreement, and halve the food waste per capita [7].

This is in line with CE impacts, and the EU has developed an action plan to promote it. The EU, thus, acknowledges this and promotes CE implementation. It dedicates funds towards it, such as the Horizon 2020 and LIFE [7], and has also dedicated a set of documents, the BRIEFs, that, in different sectors, share behaviors and technologies that influence the CE.

## 1.2. Circular Economy Key Concepts and Impacts

The CE has no strict definition. However, it is commonly understood as a "system that is restorative or regenerative by intention and design. It replaces the 'end-of-life' concept with restoration, shifts towards the use of renewable energy, eliminates the use of toxic chemicals, which impair reuse, and aims for the elimination of waste through the superior design of materials, products, systems, and, within this, business models" [8].

International actors such as the EU, China, Google, and the Ellen MacArthur Foundation (EMF) are intensifying legislation, technology, and research that promote CE. This allocates the CE into different levels and sectors, as is expected of an economic system, and its definition adapts to each one. Therefore, one can expect a different, but not contradictory, definition of a CE for different purposes, for instance from a business perspective, or a political perspective.

Today, a CE can be commonly understood as a system with a holistic impact that works in loops, at different levels, which mimic the loops seen in nature [9]. At its core there is the design for second usage, the goal to eliminate waste and to avoid toxic materials, the importance of waste management, and the implementation of the 9Rs (reduce, reuse, recycle, recover, refuse, repair, refurbish, remanufacture, and repurpose) [10].

Moreover, it is acknowledged that the outcome of the CE is a more sustainable economy, referenced as growth from within. However, to achieve it, businesses need to be transparent and cooperative [11].

This will require a more active, aware, and skilled society. Being flexible, modular, and resilient, it needs to take innovation into account [12].

CE implementation can have significant economic and environmental impacts. Through a systemic implementation of a CE, it is estimated that European gross domestic product (GDP) can increase by 7%, reflecting annual savings of 600 billion EUR, benefits of 1.8 trillion EUR per year, and the generation of 170,000 jobs by 2035 [6]. Furthermore CE can reduce carbon dioxide emission by 48% by 2030 and 83% by 2050 [12]. During the same period, primary material consumption can be reduced 32% and 53% [13].

Today, different tools for assessing circularity exist, such as the Butterfly Diagram, the RESOLVE, mass flow analysis (MFA), and life cycle analysis (LCA) [13–16]. These frameworks were not designed specifically for a CE in cities but incorporate some of its features. To date, CE research has focused on the business side and on a global perspective [2]. This framework brings the analysis to the city level, while discretizing it in sectors.

### 1.3. Circular City Definiton

Cities host 50% of the world's population. They are responsible for 85% of the total GDP, 75% of natural resources consumption, 50% of waste generation, and 60–80% of total greenhouse gas (GHG) emissions [17]. By 2050, the share of the population living in cities is expected to increase to 70%, increasing the weight of these areas in economic, environment, and social matters [18]. Therefore, there is a need to explore circularity in cities, merging these two holistic and impactful trends. With increasing urbanization, cities are an ideal location to implement circular changes, originating the circular city concept. There are some cities already embracing this challenge, such as Amsterdam [14], Glasgow [19] and Barcelona [20], contributing with case studies to the literature [21]. The research in this specific area is in progress, with a need for more case studies. Moreover, there is a need to monitor circularity in an urban context, but there is a lack of tools to do so, as well as indicators [16].

However, circular cities face now what CEs faced before: the need of a definition. This definition still has to embrace the relevant aspects of CEs, but it needs to shift them to the city perspective. This results in a merging of CE and city dynamics and fundamental structures. The city is a complex system that involves areas beyond the economy and the CE [22].

Therefore, tools that are developed or are adequate for circular business analysis—such as LCA, MFA, or RESOLVE—fail to capture the circularity of a city. They are generally too specific or too broad, since they were not developed specifically for circular cities. However, these tools reflect CEs and should partially represent the city, i.e., they can still have a role in city monitorization and a definition [22]. For instance, RESOLVE focuses, among other particularities, on the consumption of materials and energy. Consumption is still relevant in a city context but is complemented with the production, as well as the flow, of materials and possible synergies [22]. Being a source of resources, instead of a drain, is a characteristic of a circular city, as is the capacity to enable connections between sectors [22]. The resources of a city, focusing on locality, together with its demography, are relevant [22]. They transmit the city context and identity. The infrastructures of the city, concerning mobility, industry, housing, and offices determine a city's dynamics. These have a heavy impact in circular terms within the city context, and must be in the definition of a circular city. Moreover, the city should be adaptable, embracing new technologies to come (i.e., digitalization, shared mobility, renewable energy, and 3D printing, inter alia) and, again, allowing synergies and material flows [22].

Finally, it is important to point out that this new approach to monitor a city, as circular and has a whole, comes with challenges. The city is composed of sectors, which can be represented in the framework to illustrate the city. These sectors need to be monitored, as well as their interactions, in terms of circularity, translating the complexity of a multi-sectorial system, which is characteristic of a city [23]. Despite the recent efforts and progress in this area, there is still a lack of data and indicators in the different sectors and in the city as a whole for implementation. Besides the lack of circular data and standardization, these challenges are burdened with a lack of circular city case studies [22].

*1.4. Portugal and Porto Context*

The Portuguese CE action plan is aligned with EU directives. It targets parallel goals on air pollution reduction and energetic dependency, among others. Several programmes support the CE action plan implementation, such as the Innovation, Technology and CE Fund, Portugal 2020 and ECO.NOMIA. By implementing a CE, Portugal expects to reduce raw material dependency by 30%, with a gross value-added (GVA) increase of 3.3 billion EUR [24].

Porto is a northern city of Portugal and was selected for this study given its goal to be a circular city by 2030. It focuses on industrial symbioses within the Eurocities programme [25]. Symbioses are important for a CE, as they allow synergies between entities that promote waste reduction and an increase in efficiency. Industrial symbioses focus these synergies within and between industries.

Porto is geographically part of the Great Porto—responsible for 12% of national wealth—and the Metropolitan Area of Porto (AMP). The average waste generation in AMP is 588.2 kg per capita every year [26]. Finally, Porto is culturally and economically defined by different sectors, such as the wine industry, the cork industry, the sea economy, the agroindustry, furniture technology, and sustainability [27].

## 2. Materials and Methods

The development of this paper is based on a literature review, complemented by data sets and semi-structured interviews of experts on the circular economy field and on the respective sectors of a city.

During this process—which was carried out throughout the entire development of the paper—the biggest challenge faced was the gathering of data and indicators, due to the lack of support and transparency of important entities.

However, other entities were also critical to its development. For instance, literature was gathered from the EU and the EMF. The interviewed experts contributed to the lack of data and indicators, and provided a more intuitive and holistic interpretation of circularity in a city.

Finally, different data sets were used. Between them are the Eurostat, the Diário da República Electrónico (DRE), the Instituto Nacional de Estatística (INE), the PORDATA—mostly supported by INE—and IRENA data sets. Alongside them were data sets provided by private companies, namely Cork Amorim and Águas do Porto.

*2.1. Literature Review*

This paper started with the intent of understanding the impacts of CE in cities. Therefore, a literature review was carried out, involving academic papers and reports that would be related to a circular economy.

The EMF is a very active actor in the field [1,12,13,15,27]. With different reports on the subject, also in collaboration with Google [17] or the World Economic Forum [8], it described the role of a CE, its economic and environmental impacts, and the lack of research. This was supported by academic papers confirming reports and providing a more scientific insight. CE key concepts, technologies, and behaviors result from these reports.

Past case studies regarding the CE in Amsterdam [14] and Glasgow [19] were insightful in how a CE is implemented and monitored today in a city. Other cities, such as Barcelona [20], London [28], Lisbon [29], and Stockholm [30], inter alia, were also studied through other entities and academic research. Alongside it came the definition of a circular city, the tools to monitor it, the in-use indicators, and sector-specific circular technologies and synergies. After analyzing different cities, the common relevant sectors and circular city dynamics were sketched.

The political framework, together with international and national goals, is important to understand where a CE is heading and the EU's commitment towards it. Different EU action plans and directives were part of the literature review to understand this [31–33]. They were then narrowed down to Portugal's different action plans and active legislation [24,29,34–44]. Finally, the Porto Circular Economy Roadmap gave insight into the Porto context, together with regional legislation [25–27,45].

When the framework was already defined, a variety of interviews were carried out with experts of the different analyzed sectors to gain insight. These interviews are available in Appendix B. A literature review of each specific subject was carried out at the same time, with the aim to better understand each sector and to collect data and possible indicators.

### 2.2. The Circular City Analysis Framework (CCAF)

The created CCAF framework aims to keep the key concepts of a CE adapted to a city perspective and to capture the circularity of any analyzed city. It is in line with different CE interpretations as well as the circular city.

Besides the aim to analyze circularity in cities, focus was placed on the framework being simple and intuitive. This will increase understanding between the city's many agents, from municipalities, to academics and enterprises [3]. Therefore, a multi-sector analysis was adopted, representing the different sectors of a circular city.

The CCAF is composed of the Circular City Diagram (CCD), shown in Figure 1, and three tables. This display aims to picture the holistic perspective of a city, as well as its multi-sectorial aspects. The combination and labeling of the fields bring flexibility, adapting to different cities, and modularity, enabling the integration of different tools and future multi-level analysis. The sectors themselves showcase the different aspects of the circular city.

The three tables complement the CCD. Firstly, the fields table describes the fields, the relevant agents, technologies, and behaviors of these fields, its indicators, its goals, and the current situation. Secondly, the synergies table describes illustrated synergies, the sectors they compromise, the goals, and the current situation. Thirdly, the policies table lists the policies, its level (regional, national, or international), and the fields it affects, describes the policy, and presents alternatives. These tables are present in Appendix A.

The CCD is organized into three areas: the inner circle, the intermediate circle, and the outer circle. The inner circle gives CE information regarding the city, as well as the source of different businesses, materials, and energy flows. The intermediate circle focuses on the industries and sectors that characterize every city. However, it does not reflect every relevant aspect of a city. It is the outer circle's purpose to capture aspects with broader fields.

Each field is composed of one or more indicators that aim to reflect the city's circularity level. Therefore, not only indicators have to be selected but also goals. These goals are chosen by identifying realistic circularity levels within a city or go alongside a certified objective set by the EU, country, or region. For each of the indicators, the % of the completion of such goals is represented, with the completion of the bar representing the indicator measuring each goal.

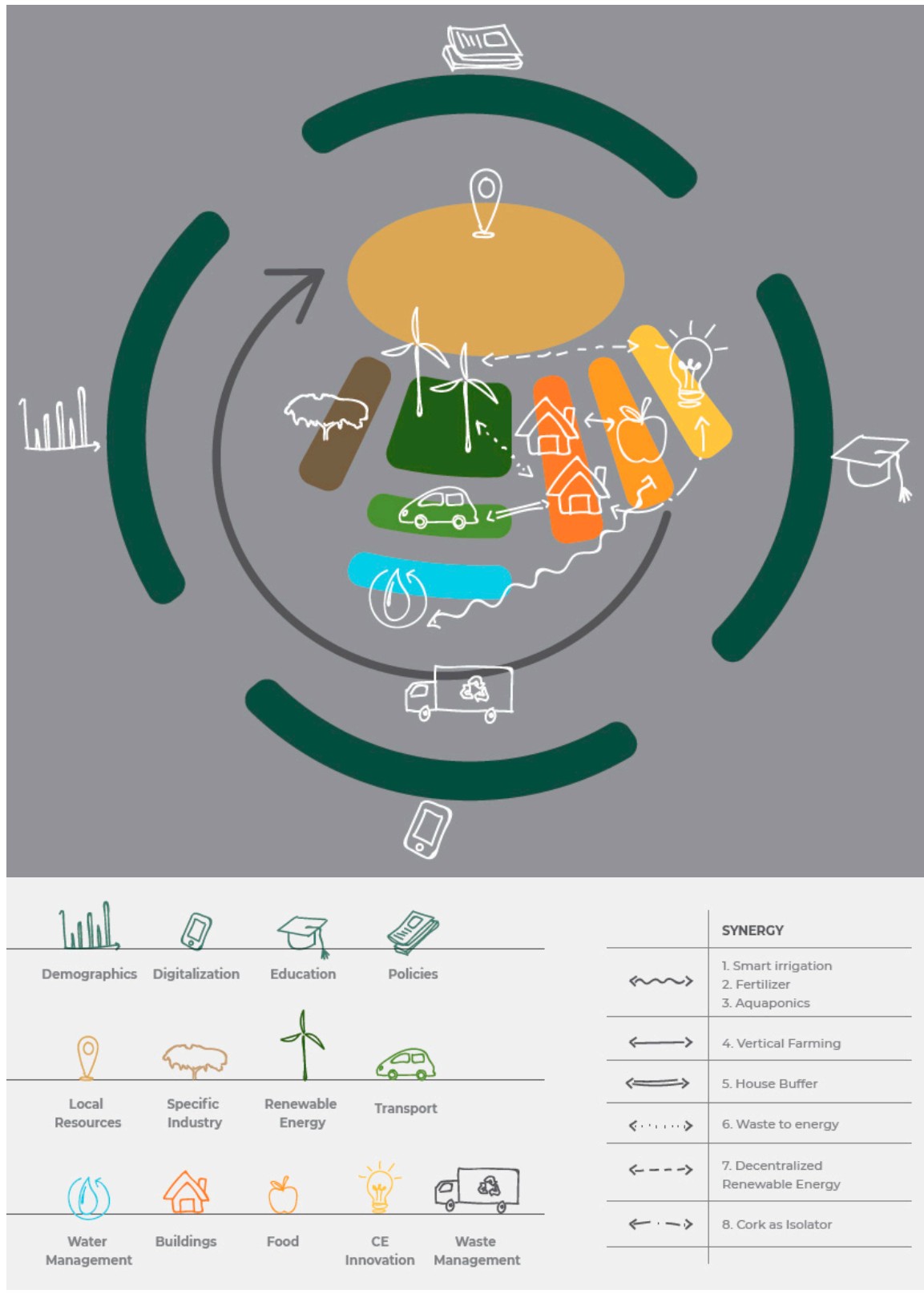

**Figure 1.** Circular City Diagram (CCD).

### 2.3. Circular Economy Indicators

Indicators are widely discussed in CE. The lack of established indicators and data are an acknowledged barrier to CE´s implementation and are currently under development. In circular cities, there is the need to define what is relevant and take into account what can be measured [46].

Linear economy indicators are no longer applicable to CEs, and this brings about the need to develop a new set of indicators to power CEs and circular cities. Despite the recent progress in this field, there is still room for improvement, especially in the standardization of indicators and data collection [47].

Academic literature is constantly proposing new indicators for CEs. Even for circular cities, some indicators are available [3]. However, generally these indicators are not applicable due to the lack of data. Moreover, different tools use different indicators that generate incomparable monitorization [48].

The CCAF aims to capture the relevant aspects of a city through the different sectors. To do so, it requires different indicators per sector as well as indicators that allow for comparison of the different sectors. Some of these indicators are available, but the data to feed them is not. Therefore, for testing in the Porto case study, a more pragmatic approach was taken, using available data and available indicators.

Hence, the Porto case study can be analyzed in its different sectors, whereby the functionality of the framework is shown, but these results cannot be considered definitive or conclusive. This means that further work must be done in this area to implement the CCAF to its full potential.

Different data sets of indicators were reviewed, and possible indicators for the sectors are already in the literature review. However, due to the innovative nature of this framework its indicators need to be tailor-made. This does not mean that other circular, sustainable, or sectorial indicators and data sets are negligible. These are present in the literature review and are the origin of the CCAF current indicators. They will also be the foundations of future indicators if further work is done in this framework.

A starting point for circular indicators is [49], where the SDGs are related to different indicators. The diverse nature of these indicators goes in line with the CE holistic impact, as well as its goals. Similar information is gathered in [46], with its proposed indicators more limited to CEs. Supporting and providing better insight into these indicators is [50], followed by [51,52], where diverse indicators are applied to EU country comparisons and United States of America (US) city comparisons, respectively.

An extensive country comparison analysis of sustainable and environmental indicators was performed in [53]. These indicators become more circular city-, business-, and economy-focused in [3,4,16].

Regardless of the relevance of this literature review, no data were available for Porto through these indicators. Furthermore, many of the indicators lacked a sectorial identity. Therefore, available data sets for Porto, generally found in INE or PORDATA, were used, leveraging on precision, but being outdated and lacking in circularity. Other indicators for Porto were considered, through sector reports and interviews. Acknowledgement of different tools such as RESOLVE, MFA, and LCA, together with city case studies [21], completed the indicator literature review.

An immense number of different indicators have been emerging in spite of the lack of data and even of their utility [4]. The CCAF requires different types of indicators. On the one hand, sector-specific indicators that capture the reality of each sector are used; on the other hand, standard indicators are used, and they apply to every city and allow for sector and city comparisons. This is a bottleneck of the framework due to the circular city indicators state-of-the-art.

### 2.4. Field-by-Field Development

To identify the circularity of the city, 13 different sectors were identified and split into three different levels: inner, intermediate, and outer circles. Fields with more possible synergies were placed closer to each other, with such synergies mostly happening in the intermediate circle.

Each field is followed by a list of indicators found in the literature review and proposed by the authors that can reflect circularity in the sector. Two sets of indicators are identified: possible indicators and used indicators. The description of the different fields is presented below:

**Inner Circle:**

- **Local resources:** The inner circle is solely composed of local resources, allowing this definition to be flexible enough to embrace different local aspects such as energy, food, material, and cultural sources. It is connected, at its core, to most of the intermediate circle sectors.

  ○　Indicators used: wind potential (m/s); solar potential ($W/m^2$); green roofs (%); imports/exports (€/€).

**Intermediate circle:**

- **Renewable energy** is in the most central position of the intermediate circle due to its overall impact. Energy connects to virtually every sector, enabling many inter-sector synergies [5]. Renewable technologies also enable waste reduction, foster efficiency, and bring a diverse and clean identity to the city and to circularity from the beginning [48].

  ○　Indicators used: renewable penetration (%); access to electricity (%); energy intensity (GWh/M€).

- **Transport sector:** A major component of renewable energy is the transport sector, playing a central role and close to different sectors. It allows synergies and is a buffer to buildings or renewable energy storage. This sector, as well as the building sector (next to the renewable energy) and the food sector (next to the building sector), is integral to every city [12]. It faces structural challenges, with cars being parked 92% of the time and only 1–2% of the total energy used to move people. Moreover, it accounts for 24.3% of GHG emissions, but can be shifted towards a more sustainable pathway through sharing systems and electrification [12].

  ○　Indicators used: public transport usage (%); electrical energy consumed in the transport sector (%).

- **Building sector:** This central sector in cities generates 14 million jobs in Europe, representing 8.8% of its GDP. It is responsible for 32% of GHG emissions and 40% of the energy consumption. This inefficient sector can be influenced by CEs, through material passports and banks, digitalization, and decentralized renewable power. For instance, 3D printing can reduce material waste and cut costs by 30% and delivery time by 50% [12].

  ○　Indicators used: retrofitting (%); very degraded buildings (%).

- **Food sector:** Besides being responsible for 40% of EU land, the food sector accounts for 19% of the European average household and 45% of the EU Commission budget. However, on average 20% of the food value is wasted through its value chain, and 11% is due to consumers [12]. Digitalization, as well as new technologies (e.g., aquaponic, urban farming, and precision farming) and circular behaviors, can transform this sector, increasing irrigation efficiency by 20–30% and reducing pesticide use by 10–20% and fertilizer consumption by 70–90% [12].

  ○　Indicators used: food waste treated (%); food waste treated in small and medium enterprises (SMEs) (%).

- **Water management** plays a central role due to its necessity. In many cities, it is an inefficient sector that can be upgraded through new monitoring technologies and smarter networks [12].

  ○　Indicators used: safe water accessibility (%); water efficiency (%).

- **Waste management** is critical to a circular society. This sector's responsibility is to collect the waste of different industries and assure for it a second life. This means planning longevity and designing waste-avoiding toxic materials, while keeping assets in the market at high value through tight loops [28]. This field limits the intermediate ring, starting in the food sector and ending in local resources.

  ○ Indicators used: landfilled waste (%); separated waste (Kg/capita*year).

- **CE innovation:** This field represents the motor of modern business creation, here with a focus towards a CE [28]. Activities such as reverse logistics can end up being incredibly complex, as can synergies [54]. An innovative business plan (or disruptive technology) can overcome this complexity, bringing about a competitive advantage for those who are implementing the CE, and enable an improved CE scenario. CE innovation is situated on the right side of the intermediate ring, close to education, due to its natural symbiosis with academic R&D.

  ○ Indicators used: CE innovation budget (%).

- **Specific industries** are on the left side of the intermediate circle. Their purpose in the framework is to highlight relevant sectors for the CE and the city. These industries are economically representative of the city. Moreover, they bring flexibility and singularity to the framework, allowing cities to monitor their different impactful sectors.

  ○ Indicators used: recycling rate (%); synergies (%).

  **Outer circle:**

- **Education** is close to CE innovation and is the backbone of every society that seeks progress. It is an important sector, since CE requires a set of skills that today societies are still lacking [1].

  ○ Indicators used: basic education quitting (%); superior course (%).

- **Digitalization** is at the bottom, influencing almost every sector. As written in [17], "[p] owering the circular economy by providing digital solutions and closing the information gap is probably the best investment that technology companies of our time can make."

  ○ Indicators used: accessibility to smartphones (%).

- **Demographics** illustrate the society of a city and, therefore, are next to specific industries that can be affected by them. These industries are representative of the characteristics and identity that make every city different.

  ○ Indicators used: balance between men & women (%); heaviest age group (years); active population (%).

- **Policies**, being on the top, represent the top-down approach of policies and reflect the legal framework in which a city is inserted.

  ○ Indicators used: man–woman balance in politics (%).

## 3. Porto Case Study

This framework was then tested in the city of Porto. The framework's indicators were quantified with the best available reports and data, combined with semi-structured interviews with experts in the field. When quantitative data were not available, only a qualitative analysis was performed. One first key result is that data paucity is an issue, and more effort will need to be dedicated to data collection.

However, the framework could help cities decide which indicators to use to start monitoring CE. The boundaries of this analysis are the Porto geographic limits, with exceptions of expansion to the AMP or Portugal.

The CCD for Porto is represented in Figure 2, with the fields (Table A1), synergies (Table A2), and policies (Table A3) tables in Appendix A. The field table lists the used indicators. Below is an explanation of the different sectors.

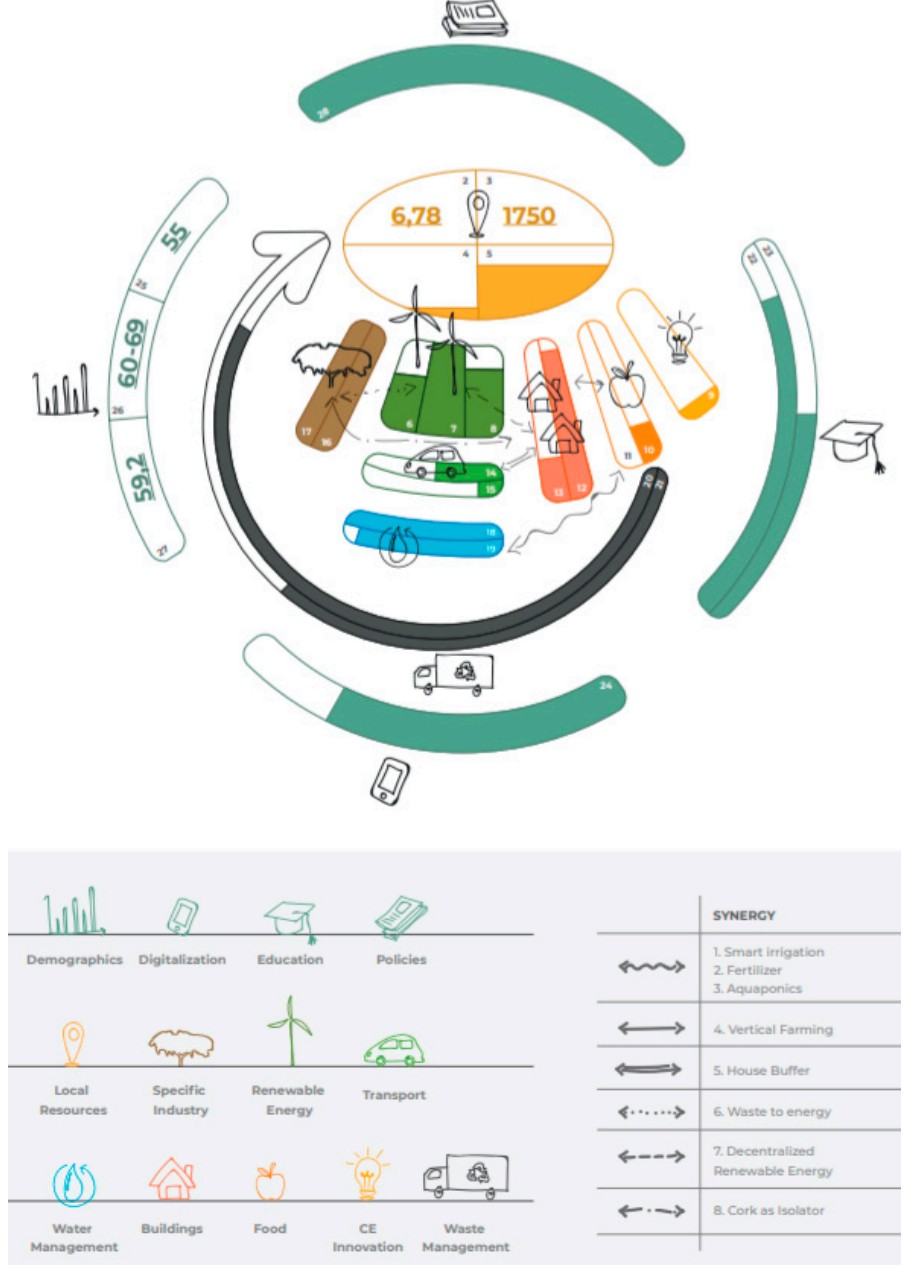

**Figure 2.** The CCD of Porto.

**Inner Circle:**

- **Local resources:** Porto has good wind and solar potential [55,56]. Porto wants to increase its green space areas, which will promote a better lifestyle and allow circular synergies, for instance, through water waste usage. Currently, it has around 0.5% of its area covered by green rooftops, an

initiative to promote green spaces [45]. Moreover, its import/export is higher than 1; in a circular city, it should be below 1, i.e., it should generate more than it consumes [57].

**Intermediate Circle:**

- **Renewable energy** is taking advantage of the wind and solar potential, representing, in Portugal, 63% of the power generated [58]. The Portuguese grid gives access to, virtually, every citizen [59]. This means that Porto is in line with the national renewable energy values. However, it has an energy intensity of 1.56 GWh/M€, above the EU average of 1.4 GWh/M€ [58].

- **CE innovation:** Porto has different innovation hubs with diverse entrepreneurial fronts. Many of these fronts can have a positive circular impact. Porto Digital, together with OPO Lab and the innovation hub, are relevant platforms that promote circularity [26]. The budget dedicated for innovation is 0.009% of the total municipal budget [60,61].

- **The food sector** is difficult to measure, due to the many stakeholders involved and the difficulty of keeping track of energy and material flow. In Portugal, Continente, a wholesale market company, embraced circular practices, making them pioneers among its competitors. Except for this market, little evidence of steps towards circularity has been found. Continente has seen positive outcomes due to its efforts in savings, sustainability, and customer engagement. Therefore, it was selected as a reference, having close to zero food waste, with the rest of the wholesale market making no contribution. This gives a share of around 21% of food waste being recovered in the wholesale market [62]. However, the SMEs in the food industry, show no evidence of circularity, and it was considered 0% [63].

- **The building sector** is representative of data paucity in the field. It indicates extreme building degradation of 1.7%, which is not the perspective Porto itself transmits. Porto's buildings are usually old, degraded, unoccupied, and not monitored. They are not prepared to embrace synergies as renewable energies. Nevertheless, there is opportunity to remodel the city, translating to a retrofitting percentage of 13.6% among all work buildings in the city [64].

- **The transport sector** does not yet indicate any positive progress. This sector aims for electrification, shared mobility, and increased usage of public transportation. A percentage of 19.6% of mobility is indeed through public transport, but electrification is blocked [65]. This can be attributed to the low incentives for this technology, alongside a monopolized charging infrastructure that is now stagnating and bottlenecking electric mobility [41]. Representative of this is the electrical energy consumed in the transport sector: only 0.6%, far below the 10% mark set by the EU [66].

- **Specific industries:** Only one was selected for Porto: the cork industry. This industry is a model of circularity in Porto. Its business relies strongly on cork as a raw material, and it is led by Cork Amorim. By verticalizing its business model, Cork Amorim expanded from raw materials to different products—from crop stoppers to space composites [67]. Cork characteristics, together with this vertical approach, allow Cork Amorim to recycle and reuse its material, allocating it for different purposes while retaining its value. Its vision and innovative perspective leads to new market opportunities for cork, bringing possible synergies with every sector of the intermediate ring of this diagram [68].

- **Water management** in Porto is an old network and lacks monitoring and nutrient extraction. It has plans in motion, oriented by Águas do Porto, to create stations to treat wastewater and generate fertilizer [12]. It features, as expected of Portugal, 100% safe water access, together with an efficiency of 81% [69]. Its efficiency is blocked by the networks and their monitorization, and upgrades to increase efficiency—at this level—are expensive and even considered economically unfeasible [26].

- **Waste management:** Lipor is responsible for waste management and is a good example in Portugal and the EU. Only 1% of the waste ends in a landfill; the rest is treated or energetically valued [70]. Lipor also organizes sensitizing campaigns, with a recent one promoting the use of

combustors by citizens, in households, generating their own fertilizer. This is reflected in a waste separation of 46.54 kg/year per capita [71].

**Outer Circle:**

- **Education** in Portugal is mandatory between the age of 6 and 18, or until the 12th grade is finished [40]. This policy promotes an educated society but has yet to translate into impactful results. Still, 11% of the students in Porto quit basic education, and only 25% of the population have good grades [72,73]. Despite the efforts to incorporate more students and to provide a better education—including circular and sustainable behaviors—there is still a lack of skills that CEs require to thrive, since they rely on a skilled labor force, which can lead a holistic, circular shift through large enterprises and SMEs [26].

- **Digitalization** is present in Portugal and Porto. Companies embrace new technologies, programming is a course gaining more spotlight, and smart metering is a discussed and aimed solution by large utilities, such as EDP and ENDESA [59]. It is a process that is already underway, combined with an equipped society and connected to the world through smartphones. In Portugal, 71.6% of the population has access to a smartphone, and this is reflected in Porto [74].

- **Demographics** represent a gender-balanced society in Porto, with up to 60% being able to work. It also translates the Portuguese aging trend, with the heaviest age group being 60–69 [75].

- **Policies**: Portugal, and consequently Porto, is in line with the EU CE action plan and consequent directives. A more social indicator was used to analyze this field, showcasing the framework adaptability to embrace different data and indicators. This also showcases the lack of data in some of the fields. The indicator consists of the percentage of women present in the municipality directive board. A cap of 30% was selected [76], and Porto overcame this with 36% [77]. In the framework, this is represented as 100%, indicating that this field has reached its goal.

## 4. Discussion

Porto is still in the initial stages of becoming a circular city. Nevertheless, the city is part of the Eurocities group and has the objective of becoming fully circular by 2030 [25]. Evaluating the city circularity with CCAF shows how the city is doing in some sectors, while still lagging in others.

The different sectors of Porto demonstrate initiative towards circularity. Most significantly, waste management, water management, and the cork industry are successful examples. CE innovation and the food and transport sectors require a critical shift towards circularity. The remaining sectors are already in a circular path and require further measures.

A higher interconnectivity between sectors—synergies that result in less waste and higher efficiency—is desirable. This can be achieved if the different sectors adopt a circular perception, together with transparency and cooperation [8].

### 4.1. Case Study Discussion

In this section, each sector will be discussed. This discussion relies mostly on Porto's performance, and lessons can be learned for other cities aiming to shift towards circularity.

Looking at Porto's local resources, the city has the potential to increase energetic independency through solar and wind. Economically, a more export-oriented philosophy must be adopted, making more use of the local resources and recirculating some of Porto's products. This is in line with the CE characteristic of being a generator instead of a consumer [15]. Finally, there is a push from Porto towards the installation of more green areas, which result in a higher life quality and enable synergies [45].

In the renewable energy field, Portugal is on a promising path, supported by a grid network that connects the entire country. A next step could be decentralized power adoption that is incentivized by the government; this would provide more energetic independence and resilience to cities and buildings.

Porto is increasing its innovation in general, with a higher focus on hubs and cooperation with universities, and benefits from a platform of entrepreneurs and innovation that promotes CEs. The actions of enrolling in international projects are a positive aspect of Porto innovation centers and will lead this transition towards a circular city.

The food sector requires major improvements, especially on the two fronts analyzed by the indicators. On one hand, the retail sector has yet to increase food waste allocated to a second life. Continente is an example in this area. On the other hand, food SMEs need to decrease their waste flow, adopting sharing schemes such as OLIO that allocate flows to citizens who want it [78].

Porto's housing must both be retrofitted and adopt circular materials and technologies. It must adopt smarter electrical networks and metering, allowing synergies from decentralized power in buildings and the integration of electrical vehicles (EVs) as buffers. Smart housing and offices should be further explored, implementing sharing behaviors and connecting through IoT [12]. Moreover, retrofit must increase its share in building works and be complemented by material passports and material banks. For these three actions to happen, legislation is required, focusing on heavier penalties for the non-registration of materials in buildings, the allocation of local banks of building materials, and a push to shift retrofitting based on circularity as a viable alternative [79].

Individual transport is the main source of transportation in Porto and Portugal, mostly composed of fossil fuel powered technologies, despite a good public transport sector [80]. This goes alongside a political panorama that only slightly promotes the shift to EVs and a very weak charging network for EVs, with its development blocked by the monopoly of MOBI.E [36,66]. To promote a shift towards public transportation, Porto can increase the area of prohibited zones for individual transportation, complemented by restrictions for fossil fuel transportation. More than just a charging network for EVs, the infrastructure of Porto needs to be rebuilt, focused on the future of mobility and making it flexible for technologies that will most likely thrive—automation and IoT, for example—in the mobility system [17].

It is understood that a shift to circularity can happen by connecting specific industries through synergies, shaping the economic and cultural panorama of the city [12]. In the case of Porto, other industries to join the cork industry may be the textile, furniture, shoemaking, plastic, rubber, metallurgic manufacturing, and wine industries [27]. These industries are all present in Porto and all share opportunities—some due to the ease of implementing circular business models, others due to the substantial potential and positive impacts that would occur if these circular business models were implemented [27]. The cork industry is a great example of how circularity can thrive in companies, with positive impacts on the environment and finances of the enterprise. The ease, compared to most other sectors, of recovering data in Cork Amorim must be highlighted. Companies such as these have the opportunity to monitor their business closely and focus on circularity, and to use it as leverage against competition. This is a sector that is expected to keep expanding, which can use the CE trend to lead the circular transition.

The Porto municipality and Águas do Porto have a high focus on water management and on how to upgrade it, especially in matters of losses [26]. The indicators used in this work do not reflect this situation. It shows the liability of current indicators, and the need for information on relevant and accessible entities. The political framework, although lacking a set of objectives and oversight, is organized to tackle the leakage issues, increase synergies by the recovery of nutrients, and upgrade the digitalization of the network [26].

The waste management sector is well developed, mostly due to Lipor. The greatest achievement is the almost zero landfilled waste, which should place Lipor as the reference to Portuguese cities. Nevertheless, many of this waste ends up incinerated, and such solution does not recover most of the value of the material. Hence, in a circular model, should be a final solution [10]. Lipor is already promoting a better second-life usage of the waste separation of materials, but it is the company's responsibility to design waste in a smart way until disassembly, using materials that are durable and circularity-friendly [13]. For that to happen, reverse logistics business models need to be explored and

supported, together with second-life material markets. Moreover, Lipor is asking the citizens of Porto (and other municipalities) to separate their waste and is educating them on how to directly reuse it at home. This is being described, by Lipor, as a success programme.

Portugal has a good political framework that promotes education. Porto follows this profile with a strong college presence, backed by some relevant schools and leads in scholar rankings [81,82]. Moreover, there is cooperation between innovation hubs and Porto University, together with the creation of a college course of circular economies. However, Portugal still has to invest in the population's basic education and the elderly, who will be a big share of the population in the next several decades. To achieve the potential of this age group as an example, education programmes focusing on CEs must be available to elder citizens.

Digitalization is the most impactful field of Porto, with the potential to impact all sectors, bringing Porto to a circularity-friendly position. It is a trend that has gained traction in past years and is supported by big enterprises— mainly Google in the circular economy—in every sector. It is reflected in population access, for instance, smartphone accessibility [17]. All sectors need to have better monitorization. The cork industry can be a role model in this matter. Allowing the gathering and treatment of data will foster more reliable indicators that will reflect the current situation of the city in circular terms more accurately.

According to policies, many incentives are in progress or planned, with the EU being responsible for many of them. Legislation must be reviewed, with a national effort to focus it on CEs. It must be accessible and organized to the public, making it easier for individuals and SMEs to understand the political framework they are navigating in while also reducing the uncertainty of investors by showing commitment to this path. The indicator of this sector, together with the demographic indicators, showcases a balanced society without gender discrimination. It is the role of Porto, Portugal, and the EU to be generally non-discriminatory, not just with respect to gender. This translates into a balanced and progressive society that will foster awareness, acceptance, transparency, and a sense of community. This can then foster circular, social, and economic development [83].

### 4.2. Framework Discussion

The framework fulfilled its purpose of supporting a structured reflection of what Porto's progress is in circularity. The relevant fields were identified, and each one was analysed individually. It provided an idea of overall progress in CE as a city.

The local resources and demographics illustrate Porto's characteristics, allowing for an understanding of the intrinsic strengths and weaknesses of the city. The other sectors reflect how far Porto is from reaching the desired CE goals.

The specific sectors successfully bring identity to the city. It demonstrates that Porto has an influential cork industry that is an example of circular business. This is an area that adapts from city to city, and with more industries analyzed in this side of the framework, the city's economic and industrial character can be better understood.

Moreover, the fields interact well between each other. They have a mix of individuality and connectivity that is ultimately showcased by the synergies. These synergies will be further explored and tested. However, it is already possible to identify possible and ongoing synergies, when complemented by the tables.

These synergies have even fewer indicators and data support than the fields though. A better design of these synergies can allow MFAs to be implemented between fields (and within fields), solidifying the connection between fields and allowing a holistic overview of the city material and energy flows. This can be achieved with further computational work and data gathering.

Data paucity is a concern for the framework and more broadly to the implementation of CE in cities. Most cities do not track circularity and are not equipped to gather this data. A good starting point would be to define the indicators, so that the data can be targeted. A framework like the one presented in this paper can support the discovery of such indicators and support monitoring efforts.

The CCAF fulfils the objective of monitoring the complexity of city circularity and setting goals. However, it misses a mapping feature, which would allow the identification of physical synergies and ease urban planning. This can be tackled with further computational work that connects the CCAF to intuitive maps, already in use in other analyses [14].

Cities such as Paris, London, Milan, and Amsterdam [23] are already taking steps towards monitoring CE in cities. The tools usually used are the MFA, the LCA, intuitive mapping, and RESOLVE [10]. Connecting the first three tools to this framework (even a simplified version that does not complement all the fields) can bring about a precise understanding of the progress of these cities towards circularity. Even more, it can bring about the standardization of tools and indicators, pushing cities to work together towards circularity and enabling comparisons.

Portugal has the opportunity, due to Porto's and Lisbon's circularity initiatives, to develop an environment where CEs in cities can thrive. For that, standard indicators that reflect each city but also allow one to compare them on a national level can be implemented. National regulators can take an important role here, making connections between cities and identifying, together with municipalities, potentials and where each can be improved. Furthermore, intercity synergies can be found through national level meetings. This can be extrapolated to the EU level.

## 5. Conclusions

Cities are a hotspot of economic, environmental, technological, and social development [20]. With an international push towards circularity in cities, a framework that can support municipalities in the pursuit of CEs is required. This work presents a framework that comes from a holistic definition of CEs in a city context, and the framework was developed to be modular, flexible, and transparent. The framework strives to represent the most relevant fields in a circular city and their interactions. It was developed to be modular enough to be applied in different areas and at different regional scales.

For its improvement, the framework should be applied to new cases and could be upgraded in several ways. A set of standard indicators that could be used in all cities should be set, and data should be collected for it. The framework could be coupled with multi-criterion analysis to reflect the weights of different indicators in each sector and enable city comparisons. It also aims to be flexible so that different levels of analysis can occur. LCA and MFA of companies from different sectors can be an extension of this framework, reaching a meso level. This can then be repeated at the product level, reaching a multi-level analysis characteristic of CEs, achieving the holistic requirement of its implementation [84].

As stated before, the target users of this framework are municipalities, alongside other agents promoting and monitoring circularity in a city. It is understood that the framework is intuitive enough to be used by non-specialist and non-scientific personnel, leading to a reflection of a city's circularity from a multi-sectorial perspective. Furthermore, matching the CCAF with an intuitive city map can create an even better understanding of where to act, who is acting, and the local synergies, and this can ease urban planning at the same time.

This study could be enriched by (and can inform) the work being done by the European Commission, together with relevant entities in the CE subject, such as EMF and ESAC, as they identify indicators to monitor circularity. While all cities are different, some standard aspects exist. Therefore, modularity and flexibility need to be present in the development of indicators. They need to be balanced with a standardization and simplicity perspective that allows for city comparisons and rankings. Consequently, the integration of multi-criterion analysis indicators is achieved, and the circularity of a city is concisely reflected.

Finally, there is a need, in terms of the analysis of a city in its circularity, for a broader involvement of agents, reducing the needed assumptions and therefore allowing a reflective understanding of a city standpoint as well as its future objectives and pathways. Circularity in a city needs to be harmonised and understood together with other action on decarbonisation [85,86]. With a broad enough work

group, deeper analysis can become a reality, as this framework with economic, technical, social, and environmental analyses of possible paths will be complemented.

**Author Contributions:** This article was co-developed by the two authors. A.C.F. developed the core of the paper and underlying analysis and F.F.N. supervised the work and co-wrote & reviewed the paper.

**Funding:** The work presented in this paper was funded by the European Commission under contract number no. 642242 (https://deeds.eu/).

**Acknowledgments:** The graphical design of the framework was developed together with Teresa Segismundo. English revision was kindly provided by David Hughes and Charlotte Wragg.

**Conflicts of Interest:** The authors declare no conflict of interest.

## Appendix A  Framework Tables, for Porto

To complement the framework, three tables were developed. These tables work as the backbone of the CCD and provide more insight to each sector. They are the starting point to develop the CCAF, since it is in there that most of the information is stored. However, in this case study, it is important to remember the lack of data and support. This led to less insightful tables. Nevertheless, as well as the CCD, the tables need to be updated once more recent and reliable information is gathered.

The first table analyzes each field in more depth. It is divided into different columns, with the left column listing the different fields. First, it provides a description of each field, contextualizing the results and providing qualitative informat ion that cannot be reflected by the indicators. Second, it identifies the principal agents influencing that sector. This can be extremely useful if the CCAF is merged with an intuitive map, since it indicates which entities should be highlighted. Third, it lists the technologies in use in each field that promote circularity. This column can be further explored by matching it with a list of all possible technologies and behaviors that can promote CEs. Again, this is useful when merging CCAF with other tools, because it determines the context of that field. The last three columns refer to the indicators that are in use, the current value of that indicator, and the desired goal.

The second table analyzes the synergies in action in the city. This table is extremely useful, since an extensive visualization of these synergies in the CCD would increase the framework complexity and hinder its understanding. As a side note, synergies could be further explored in the CCD if this framework is adapted into a computational framework. After listing the different synergies in the first column, the second column indicates the fields that the synergy involves. The next column describes the synergy itself and contextualizes it in the analyzed city (in this case, Porto). The following two columns describe the current situation of the synergy, followed by the aimed goal. These two columns can be further explored, as they apply quantitative goals to the different synergies as well as indicators.

The third table focus on policies. Policies have a heavy impact on city dynamics and its future. Moreover, due to its holistic and qualitative characteristics, they are difficult to translate into indicators in the CCD. Hence, this table complements the policies field. It lists different policies, on different levels, that impact and define the city circularity. Mirroring the two other tables, the policies are listed in the first column. This is followed by a column identifying the level of implementation of the policy (regional, national, EU, global, etc.) and then the fields affected by it, similar to the synergies table. Another column describes the most relevant points of the policy, followed by another with recommendations on that policy, so that it better aligns with the CE objectives and context of the city.

**Table A1.** Fields table.

| Field | Description | Agents | Technologies/Behaviors | Indicator | Current | Goals |
|---|---|---|---|---|---|---|
| **Local Resources** | Source of energy in Porto, its macro-economic profile and life quality | AMP, Câmara Municipal do Porto | Green spaces, energy source data, air pollution levels | 1. Wind Potential (m/s) | 6.78 | - |
| | | | | 2. Solar Potential (W/m$^2$) | 1750 | - |
| | | | | 3. Green Roofs (%) | 0.5 | 10 |
| | | | | 4. Imports/Exports (€/€) | 1.5 | 1 |
| **Renewable Energy** | A broad analysis considering the Portuguese grid and local production | EDP, REN, Endesa, DGEG | Decentralized production, PVs, wind kit-based, biomass, waste-to-energy | 5. Renewable Penetration (%) | 63 | 100 |
| | | | | 6. Access to Electricity (%) | 100 | 100 |
| | | | | 7. Energy Intensity (GWh/M€) | 1.56 | 1.4 |
| **CE Innovation** | Platforms and business that lead to innovation in CE subjects | Innovation Hub, OPO Lab, ScaleUp, Porto Digital | Platforms connecting academics, companies and entrepreneurs, public incentives, hubs | 8. CE Innovation Budget (%) | 0.009 | 0.5 |
| **Food** | Food value chain focused on retailers and SMEs embracing urban production | Continente, Canal Horeca, Pingo Doce, Intermarche | Aquaponics, hydroponics, urban and peri urban farming, smart irrigation; vertical and community farming | 9. Food Waste Treated (%) | 21 | 100 |
| | | | | 10. Food Waste Treated in SMEs (%) | 0 | 30 |
| **Buildings** | Buildings profile, relating housing and abandoned buildings | Câmara Municipal do Porto, OASRN | Housing sharing, office sharing, retrofitting, 3D printing, industrial building work, material passport, bank of materials | 11. Retrofitting (%) | 13.6 | 50 |
| | | | | 12. Very Degraded Buildings (%) | 1.7 | 0 |
| **Transport** | The mobility within Porto, regarding the shift towards EM | MOBI.E, STCP, UBER, Endesa | Shared mobility, smart transport infrastructure, EVs, automation | 13. Public Transport Usage (%) | 19.6 | 50 |
| | | | | 14. Electrical Energy Consumed in the Transport Sector (%) | 0.6 | 10 |
| **Specific Industry—Cork** | Overview of the cork industry labeled as circular and a world leader in its area | Amorim | Cork composites, recycling | 15. Recycling Rate (%) | 100 | 100 |
| | | | | 16. Synergies (%) | 100 | 100 |
| **Water Management** | Water issues regarding its treatment and distribution | Águas do Porto | Nutrients recuperation, leakage monitoring, recirculation | 17. Safe Water Accessibility (%) | 100 | 100 |
| | | | | 18. Water Efficiency (%) | 81 | 85 |
| **Waste Management** | Recovery and treatment of waste generated in Porto, as well as the actions of the principal agents | Lipor | Ecopontos, house waste treatment, incineration, digitalization of the separation system | 19. Landfilled Waste (%) | 1 | 0 |
| | | | | 20. Separated Waste (Kg/capita*year) | 46.54 | 70 |

**Table A1.** *Cont.*

| Field | Description | Agents | Technologies/Behaviors | Indicator | Current | Goals |
|---|---|---|---|---|---|---|
| **Education** | Levels of overall education in Porto, including college and its embracing of CE | Ministério da Educação, Câmara Municipal do Porto | CE Schools, programmes in universities, sensitizing projects to overall citizens | 21. Basic Education Quitting (%) | 11 | 0 |
| | | | | 22. Superior Course (%) | 25 | 50 |
| **Digitalization** | The digital overview of citizens combined with the digital platforms and infrastructures that lead to CE | Google, INESC, EDP, REN, Endesa | Smart metering, asset tagging, geospatial information, big data management, connectivity | 23. Accessibility to Smartphones (%) | 71.6 | 100 |
| **Demographics** | The demographic profile of Porto, showing weakness and potentials | INE | Main data collection from Censos | 24. Balance between Men & Women (%) | 55 | - |
| | | | | 25. Heaviest Age Group (years) | 60–69 | - |
| **Policies** | An overview of the commitment of Porto political environment towards CE | Governo de Portugal, EU, EC, Câmara Municipal do Porto, AMP | Incentives, tax penalties, transparency, municipalities autonomy | 26. Active Population (%) | 59.2 | - |
| | | | | 27. Man–Woman Balance in Politics (%) | 38 | >30 |

**Table A2.** Synergies table.

| Synergy | Fields | Description | Current | Goals |
|---|---|---|---|---|
| **Smart Irrigation** | F + WatM | Usage of the wastewater as input to irrigation system | Already implemented, but small scale | After feasibility study, if positive, increase the smart irrigation network |
| **Fertilizer** | F + WatM | Collection of nutrients from wastewater and transformation into fertilizer for food production | Águas do Porto investing to make it real | Extract most of phosphorus and cellulose fiber, reusing it in fertilizers, reducing dependency on it |
| **Aquaponics** | F + WatM | Combination of water treatment and food production through aquaponics | No implementation yet, only referred as a possibility | Exponential growth, using wastewater and the river to develop fisheries and food production and to tackle water waste |
| **Vertical Farming** | B + F | Implementation of vertical farms in unused building areas | Close to zero presence | Citizen taking care of this business model, exploring self-production and community farms |
| **House Buffer** | T + B | Remodeling of house systems to include EVs as buffers and ESS. | Still to implement due to lack of technology and infrastructure | Product available in the market and leveraged by ESCOs |
| **Cork as Isolator** | C + B | Usage of waste cork as wall and floor isolation (sound and heat) | Already part of Amorim strong ramifications of business | Besides other synergies to Amorim, more companies to follow the lead in this type of synergies |
| **Decentralized Renewable Energy** | B + RE | Leverage building heights to gather solar, wind, or rain energy through decentralized technologies | Installations too small to be considered | The citizen embraces self-production, buildings, in conditions, having at least one decentralized technology promoting sustainability |

**Table A3.** Policies table.

| Policies | Level | Fields | Description | Recommendation |
|---|---|---|---|---|
| **Circular Economy Action Plan (CEAP)** | EU | All | Mainstream CE; showcase CE impacts; creation of second-life market for products | |
| **Waste to Energy** | EU | WasM | Waste role in CE and EU; waste hierarchy; financial supports; recommended technologies | |
| **Legislative proposal on Online Sales of Goods** | EU | WasM; D | Protection of the customer in online sales; assurance of longevity of products; promotion of reuse | Complement with digital possibilities to the market creation |
| **Legislative Proposal on Fertilizers** | EU | F; WatM | Creation of second market for recovered nutrients | Focus on local markets |
| **Directive on the Restriction of the Use of Certain Hazardous Substances in Electrical & Electronic Equipment** | EU | WasM | Creation of second-life market for WEEE; substitution of hazardous components in electrical and electronic equipment | Focus on local markets |
| **CE Package** | EU | All | Set of several goals: 65% of municipal waste prepared for reuse/recycle and less than 10% to landfill, 75% of packaging prepared to reduce/recycle, reduce maritime litter by 30%, halve global food waste in retailers and consumers, by 2030; define priority sectors; discuss monitorization | Discuss city standardization, focus on indicators and available data |
| **Programa Casa Eficiente** | Portugal | B; D | Subsidies (100 M€ EIB + 100 M€ others) to retrofit buildings in efficiency; supported by digital platform | |
| **Decreto-Lei No. 46/2008** | Portugal | B; WasM | Discretization of waste in the building sector; barriers; role of municipalities; discretization of fiscal penalties | More legislation to allocate a bank of materials; heavier fiscal penalizations if non-registration of materials used in buildings |
| **Lei No. 10/2014** | Portugal | WatM | Showcase of Portuguese national water system; entities involved and how to regulate | Set of objectives and fiscal incentives/penalties |
| **Decreto-Lei No. 141/2010** | Portugal | R | Set of different goals to achieve by 2020: renewable share in energy consumption of 31% and increase by 10% in transports, reduce energy dependency by 74%; impact of this implementations; role of municipalities and autonomy in renewable energy | strategy for different technologies; incentives for decentralized production; citizen assessment |
| **Decreto-Lei No. 82D/2014** | Portugal | T | Fiscal benefits for ICE alternatives; bike-sharing implementation | |
| **PNAEE 2017–2020** | Portugal | All | Increase co-generation production; reduce energy consumption in buildings by 1.5%; increase fast-charging stations for EVs; interest in EVs, scooters, bike-sharing; interest in renewables sources | focus on decentralized production; discuss monitoring and upgrade of metering; discuss charging stations for EVs |

## Appendix B  Table of Interviews

This paper was complemented by a set of interviews, listed in Table A4. These semi-structured interviews were always desired for the development of the framework.

Due to the holistic characteristic of CEs and circular cities, alongside the lack of indicators, data, and conceptualization, it was understood that insight from experts in different fields of CEs, sustainability, and sectors influencing a city would contribute positively to the framework.

The interviewed experts come from different sectors.  However, each one has holistic understandings of the CE impact, certain opinions on the circular city, and expertise in their sector, as well as insight in other relevant sectors.

The interviews, independent of duration and type, started with particular questions regarding the position of the expert. They then followed the same procedure as per the literature review. After obtaining inputs of the different experts' conceptualization of CEs, their experience with it, how their sector and business contemplated circularity, and which policies were more influential, it was asked what indicators they could provide and recommend to monitor circularity—in a city, in their sector, and in their business.

The interview began with brainstorming.  The future of each sector was discussed, different technologies and trends were analyzed, and the future circular city was considered.  Insights were extremely useful due to the openness of these specific experts. They shared their knowledge and their opinions and expectations for a circular economy.

At one moment of the interview (but always during the brainstorming), the framework was presented. Here, it was tested: Questions were asked regarding its adaptability to different cities and how well the framework captured the circularity of a city. Finally, it was contextualized in the city of Porto. This part of the interview helped position the different fields in CCD and define their labels.

All the interviews contributed with powerful impacts. It is understood that, in such a holistic framework and analysis, a broader group of experts should be interviewed. The implementation of the Delphi method has been suggested in the future, with a broader group of interviewed experts [4].

**Table A4.** List of interviews.

| Name | Position–Company | Interview Type/Duration |
| --- | --- | --- |
| **Elsa Rodrigues Monteiro** | Head of Sustainability and Corporate Communication–Sonae Sierra | Face-to-Face/1 h |
| **Diana Nicolau** | Marketing, Education and Comunication Technician–Lipor | Distance/1 h |
| **Joana Sousa Lara** | Co-founder–Panana | Distance/1 h |
| **Pedro Vieira e Moreira** | Head of IT & Innovation–Águas do Porto | Distance/0:30 h |
| **Vítor Martins** | Head of Environmet–Modelo/Continente Supermarkets, S.A. | Distance/1 h |
| **Nuno Ribeiro da Silva** | Portugal Director & Invited Professor of Lisbon University–ENDESA | Face-to-Face/2 h |
| **Pedro Pinto** | Business Development & Franchise Director–Cooltra | Face-to-Face/1 h |

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
