# Peer review of "A Framework for Implementing and Tracking Circular Economy in Cities: The Case of Porto"

_sustainability, doi:10.3390/su11061813_

Round 1

Reviewer 1 Report

Please find attached all my comments and suggestions.

Author Response

The authors took high attention to your remarks. These were adressed and are presented in the document below, 'Reviewer1'.

The paper itself suffered major restructure and editing. Therefore, let the authors know if you want to check the new version.

Best Regards and thank you once more,

António Cavaleiro de Ferreira

Reviewer 2 Report

The proposed research « A Framework for Implementing and Tracking Circular Economy in Cities: The Case of Porto» falls within the scope of Sustainability. According to the reviewer’s opinion, major revisions are required in order to accept this research study for publication in Sustainability. Please, comply with the following suggestions and comments:

Comment 1: In my opinion, the aforementioned manuscript needs more data in order to be published in Sustainability. It approaches the subject very theoretically. I am not so sure if your readers will find innovative data in this work.

Comment 2: In my opinion, the text needs better formatting in several parts.

Comment 3: The question that it needs to be answered is how it extends the existing knowledge on the topic.

Comment 4: When you submit the corrected version, please do check thoroughly, in order to avoid grammar flaws.

Author Response

The authors took high attention to your remarks. These were adressed and are presented in the document below, 'Reviewer2'.

The paper itself suffered major restructure and editing. Therefore, let the authors know if you want to check the new version.

Best Regards and thank you once more,

António Cavaleiro de Ferreira

Reviewer 3 Report

Dear authors. When reviewing the text I was very expectant about the methodology used, as well as the conclusions. The work is interesting, however, I think it is necessary to deepen the methodology used, for example, specify how the indicators were chosen, based on what criteria. There is no information regarding where the information was obtained.

Although the synergies shown in Figure 1 are true, they are presented in more depth in the annexes. I believe that by not indicating in the main text what they mean. The lack of explanation does not complete the idea that we want to expose in the Figure.

Just stating that interviews were conducted for my point of view is not enough to expose the results. It is not indicated who was interviewed, what type of questioning was made to the interviewees or the number of participants.

How the indicators were chosen, as well as where the information was obtained to apply them to the city of analysis.

The main text indicates that more detail would have in the Annexes, but being several tables and not indicated in the main text to which they refer, makes it very difficult to follow the idea.

It is important to check if the citations are properly placed in the bibliography.

For my point of view, properly presenting the methodology makes an article suitable and then replicating it. In this manuscript the methodology is not clear, likewise there is no discussion that partially confronts the results with other cities. There is a lack of a bibliographic analysis that allows to discuss the results obtained.

Author Response

The authors took into high consideration your comments. These are answered in the file attached below, 'Reviewer3'.

We would like to highlight that the paper itself suffered major restructure and editing. Therefore, if you want an updated version let us know.

Best Regards,

António Cavaleiro de Ferreira

Reviewer 4 Report

The paper deals with the theme of the Circular Economy applied to an urban context with the intention of designing and testing a framework for the evaluation of the circularity of cities, applying it to a case study.  The topic is widely discussed in the literature although, as the authors rightly point out, there is a lack of adequate tools for the identification of circularity indicators and their measurement. Therefore, the focus of the study would be potentially interesting, but in this form the paper should not be published. The reasons for this inadequacy for publication are described below by points.

1.Does the introduction provide sufficient background and include all relevant references?

The background provided is insufficient to introduce and delimit the subject of the study. Many of the major and most recent studies on circular cities are omitted, for example:

Circular economy in cities: Reviewing how environmental research aligns with local practices

https://doi.org/10.1016/j.jclepro.2018.05.281

Urban Circular Economy: The New Frontier for European Cities' Sustainable Development

https://doi.org/10.1016/B978-0-12-813964-6.00012-4

Circular cities

https://doi.org/10.1177/0042098018806133

Circular Cities: Challenges to Implementing Looping Actions

https://doi.org/10.3390/su11020423

2.Is the research design appropriate?

The authors state in lines 49-54 that after an analysis of what characterizes circularity in a city, they will develop a reference framework to be applied to the case study of Porto. However, I found no trace of the conceptualization of the meaning of "circular city", probably because there is no effective analysis of the literature. Therefore, the reader does not understand how the framework of analysis that the authors propose, has been built. 

3.Are the methods adequately described?

From a methodological point of view, starting from line 85, the authors argue that the framework is drawn from case studies (which are not mentioned in this part of the paper) and from key policies at macro (EU), meso (nation) and micro (city) levels.  Later it is stated that the analysis of the literature has allowed to identify the relevant fields that were used to build the framework, but there is no trace of this analysis. Therefore, the paper is methodologically poor because the framework, which is the main result of the study, is not sufficiently correlated with a theoretical basis that strengthens its foundation.

4.Are the results clearly presented?

Unfortunately, the results are limited to the mere description of the level of sustainability performance achieved by the city of Porto. For example, it is not clear what advantages the application of the model has brought. The holistic approach is not sufficient to validate its effectiveness and justify its use, as this approach is the basis of cycle tools, such as LCSA (Life Cycle Sustainability Assessment), which are already widely used to assess environmental and socio-economic impacts within the circular economy. 

5.Are the conclusions supported by the results?

Throughout the paper and in particular in the conclusions, the regenerative vision underlying the circular economy is not sufficiently highlighted. It would have been interesting to deepen how this vision applied to the urban environment, can be contrasted with the traditional linear model, just leveraging on the well-known principle of the 5R: Reduce, Reuse, Recycle, Repair, Refuse. How do the description of the sustainability results achieved by the city of Porto relate to this new paradigm? All this is not well explained.

Author Response

The authors took high attention to your remarks. These were adressed and are presented in the document below, 'Reviewer4'.

The paper itself suffered major restructure and editing. Therefore, let the authors know if you want to check the new version.

Best Regards and thank you once more,

António Cavaleiro de Ferreira

Round 2

Reviewer 1 Report

The authors addressed seriously and rigorously the numerous comments mentioned by the three/four reviewers (the cover letter addressing my comments was clear and relevant), and several parts have been rewritten and/or restructured, resulting in a much better underpinned manuscript. I have highly appreciated the modifications and improvements performed on the present manuscript.

Particularly, (i) the contributions and key locks solved in the article are explicitly stated; (ii) the introduction has been completely restructured and improved; (iii) same for the literature background part which is now more complete/consistent; (iv) the materials and methods section has been significantly improved, which was a real good point compared to the initial submitted version.

I have only some minor suggestions:

l. 76: avoid statement like "CE is an open philosophy"

l. 89: maybe explicit the "9R"

l. 133: "inter alia" in italic

l. 147-149: "CE action plan" "CE Action plan", please ensure consistency

sub-section 2.3: would prefer "Circular economy Indicators" instead of "Indicators"

l. 302: "W/m^2" please use proper mathematical notation

l.639: "This study could be enriched by (and enrich)" ? maybe "enriches"

Please cite properly and entirely the references, e.g. [3] (line 743), [6] (line 748), etc.

Overall, I think this article should be accepted for publication, after being checked by a native-english editor to ensure the diction, language grammar and English spelling, and after checking the completeness of the references.

Author Response

Dear reviewer,

Thank you for your comments. They were taken into account and a broader explanation is attached.

The authors believe the bibliography matches the standarts, and mendeley was used.

However, if you disagree we would like extra time and if possible some guidance in how to improve this part of the paper.

Best Regards,

António Cavaleiro de Ferreira

Reviewer 2 Report

The authors have complied with my suggestions. Therefore the paper should be accepted for publication in its current form.

Author Response

Dear reviewer,

Thank you for your comments. They were taken into account and a broader explanation is attached.

Best Regards,

António Cavaleiro de Ferreira

Reviewer 3 Report

Dear authors

I have reviewed the corrections and I appreciate that my suggestions have been taken into account. I consider that the document has been improved and my doubts mentioned in the previous revision have been solved.

Author Response

(The authors gave the same response as above.)

Reviewer 4 Report

Dear Authors,

I congratulate you on your significant work in improving your paper. The additions you have made by incorporating the indications of the reviewers have certainly valorised your research and made it easier to read. Therefore, in this latest version I consider your paper suitable for publication.

Kind regards

Author Response

(The authors gave the same response as above.)
